# Molecular Mechanisms of Reversal of Multidrug Resistance in Breast Cancer by Inhibition of P-gp by Cytisine N-Isoflavones Derivatives Explored Through Network Pharmacology, Molecular Docking, and Molecular Dynamics

**DOI:** 10.3390/ijms26083813

**Published:** 2025-04-17

**Authors:** Chuangchuang Xiao, Xiaoying Yin, Rui Xi, Chunping Yuan, Yangsheng Ou

**Affiliations:** 1College of Chemistry and Chemical Engineering, Shanghai University of Engineering Science, Shanghai 201620, China; xiaochaung8560@163.com (C.X.); cereliajia@foxmail.com (R.X.); 34210005@sues.edu.cn (C.Y.); 2Shanghai Frontiers Science Research Center for Druggability of Cardiovascular Noncoding RNA, Institute for Frontier Medical Technology, Shanghai University of Engineering Science, Shanghai 201620, China; 18716066360@163.com; 3Shanghai Engineering Research Center for Pharmaceutical Intelligent Equipment, Shanghai University of Engineering Science, Shanghai 201620, China

**Keywords:** cytisine N-isoflavones derivatives, breast cancer multidrug resistance, network pharmacology, molecular docking, molecular dynamics simulation, P-gp

## Abstract

The compound CNI1, identified as a novel antitumor agent based on the cytisine N-isoflavones scaffold, and its series of cytisine N-isoflavones derivatives (CNI2, CNI3, and CNI4), were first isolated from bitter bean seeds, a traditional Chinese medicinal source, by our research team. Cellular activity assays combined with virtual screening targeting P-gp revealed that CNI1, along with the three cytisine N-isoflavones derivatives, CNI2, CNI3, and CNI4, exhibited significant multidrug resistance (MDR) reversal activity in breast cancer. Despite this promising outcome, the precise molecular mechanisms and key targets involved in the MDR reversal of these compounds remain to be elucidated. To explore potential mechanisms, targets for CNI1, CNII2, CNI3, and CNI4 (CNI1-4) were predicted using SwissTargetPrediction and Pharmmapper databases, while MDR-related targets in breast cancer were retrieved from OMIM and GeneCards. The overlapping targets were utilized to construct a protein–protein interaction (PPI) network to identify core targets. Additionally, Gene Ontology (GO) and Kyoto Encyclopedia of Genes and Genomes (KEGG) enrichment analyses were conducted using the DAVID database to identify relevant signaling pathways. Molecular docking simulations were employed to evaluate the binding sites and energies of CNI1-4 with the identified key targets, with the highest binding energy complexes selected for subsequent molecular dynamics simulations. This study identified 81 intersecting multidrug resistance (MDR) targets and 19 core targets in breast cancer. GO and KEGG pathway enrichment analyses revealed that MDR was primarily mediated by genes involved in cellular processes, apoptosis, protein phosphorylation, as well as the MAPK and PI3K-Akt signaling pathways. Molecular docking studies demonstrated that the binding energies of P-gp, AKT1, and SRC to CNI1-4 were all lower than −10 kcal/mol, indicating strong binding affinities. Molecular dynamics simulations further confirmed the stable and favorable binding interactions of CNI1-4 with AKT1 and P-gp. This study provides preliminary insights into the potential targets and molecular mechanisms of cytisine N-isoflavones compounds in reversing MDR in breast cancer, offering crucial data for the pharmacological investigation of CNI1-4 and supporting the development of P-gp inhibitors.

## 1. Introduction

Breast cancer, the most common malignancy among women, remains a leading cause of cancer-related morbidity. Chemotherapy is a primary therapeutic approach; however, prolonged treatment often results in the emergence of multidrug resistance (MDR) in tumor cells, which significantly diminishes therapeutic efficacy. MDR is responsible for approximately 90% of cancer-related deaths [1,2,3,4]. Extensive research has established that ABC transporters, particularly P-gp/ABCB1, play a pivotal role in multidrug resistance (MDR) by actively pumping chemotherapeutic agents out of cancer cells, leading to significantly reduced intracellular drug accumulation [5,6,7,8,9,10,11]. Although multiple generations of P-gp inhibitors (including verapamil and its structural analogs) have been developed, their clinical application has been limited by two major challenges: dose-limiting toxicity and unsatisfactory therapeutic efficacy [12,13,14,15,16,17]. These limitations highlight the critical need for developing next-generation P-gp modulators that exhibit both enhanced anticancer activity and minimized systemic toxicity [18].

The novel natural compound, cytisine N-methylene-(*4′,7*-dihydroxy-*3′*-methoxy)-isoflavone, was isolated for the first time from the Chinese herb bitter pea. This compound, which exhibits significant antitumor activity [19], served as the basis for further structure optimization. A series of cytisine N-isoflavones derivatives (CNI2-30) were synthesized and subjected to virtual screening against P-gp targets, as well as cellular activity assays. Among these derivatives, CNI1-4 (comprising CNI1, CNI2, CNI3, and CNI4) demonstrated the ability to notably enhance the sensitivity of adriamycin (DOX) and effectively reverse DOX resistance. Of these, CNI3 showed the most potent reversal effect on MCF-7/DOX-resistant cells in combination with DOX, achieving a 9.39-fold reversal multiplicity and reducing the IC50 of adriamycin from 80.48 μM to 9.53 μM (experimental data in Appendix A). However, the specific targets and molecular mechanisms underlying the reversal of adriamycin resistance by these novel compounds remain unclear and warrant further investigation.

Elucidating drug mechanisms requires a multidisciplinary approach that combines both experimental and computational methods. While conventional techniques—including cellular models, animal studies [20,21], and multi-omics analyses (genomics, proteomics, and metabolomics) [22,23,24]—provide reliable data, they often involve substantial costs and time investments. To overcome these limitations, computational approaches such as network pharmacology, molecular docking, and molecular dynamics simulations have emerged as powerful alternatives that enable the systematic, multi-level investigation of drug mechanisms with significantly reduced experimental expenditures. This integrated computational strategy has proven particularly valuable in anticancer research. For instance, Jiao and Shi et al. successfully applied network pharmacology combined with molecular docking and dynamics simulations to reveal the therapeutic mechanisms of *Scutellaria baicalensis* in breast cancer [25]. Similarly, Chen and Liu et al. employed this approach to delineate the anti-leukemic mechanisms of 1,4-naphthoquinone derivatives [26]. However, despite these advances, the molecular targets and precise mechanisms of action for our newly discovered MDR-reversing agent CNI1-4 remain unclear.

To bridge this knowledge gap, we established a comprehensive computational framework (Figure 1) consisting of three key components: (1) target prediction using SwissTargetPrediction (version 2019) and PharmMapper (version 2017), followed by intersection analysis with established breast cancer MDR targets from the OMIM and GeneCards databases; (2) the identification of core targets through protein–protein interaction network analysis coupled with functional enrichment (GO/KEGG) using DAVID (version 6.8) (3) the evaluation of binding modes via molecular docking, with subsequent molecular dynamics simulations to validate the stability of high-affinity complexes. This multi-dimensional strategy not only provides systematic insights into CNI1-4’s MDR reversal mechanisms but also establishes a crucial theoretical foundation for developing next-generation P-gp inhibitors with improved efficacy and safety profiles.

## 2. Results and Discussion

Our team initially isolated a novel natural compound, chrysoerioline N-methylene-(*4’,7*-dihydroxy-*3’*-methoxy)-isoflavone, from the Chinese herb bitter pea. Based on this, a series of chrysoerioline–isoflavone derivatives were synthesized, leading to the identification of four optimal compounds (CNI1-4) for reversing drug resistance through structural optimization [19]. The molecular structures of CNI1-4, along with the structural formula of the positive control VRP, are displayed in Figure 2.

### 2.1. Prediction of Potential Targets for CNI1-4 to Reverse Multidrug Resistance in Breast Cancer

A total of 160 potential CNI1-4-interacting targets were identified from the Pharmmapper (version 2017) and SwissTargetPrediction (version 2019) databases, while 2618 proteins associated with breast cancer MDR were retrieved from the Genecards and OMIM databases. Cross-referencing these sets yielded 81 intersecting targets, as illustrated in the Venn diagram (Figure 3A). The left side of the diagram highlights CNI1-4-interacting targets (blue), the right side represents breast cancer MDR-related targets (yellow), and the overlapping region denotes the shared targets.

### 2.2. Construction of PPI Network

The potential targets of CNI1-4 for reversing MDR in breast cancer were submitted to the STRING (version 12.0) database, with “Homo sapiens” selected as the species and a combined score threshold set at >0.7 for data import. The PPI network was subsequently constructed using Cytoscape 3.9.0, with the results presented in Figure 3B. The network comprises 781 edges and 81 nodes, with node color variations indicating the strength of the interactions. The average values for three metrics—degree, betweenness, and closeness—were calculated, and targets with higher degree values were identified as core targets. Figure 3C displays the top 10 core targets, with protein kinase B (AKT1) exhibiting the highest degree value of 55. Figure 3D illustrates the network analysis of these core targets. The analysis identified AKT1, ALB, HSP90AA1, EGFR, CASP3, ESR1, CCND1, SRC, PPARG, and MAPK1 as key MDR targets in breast cancer. Notably, AKT1 was involved in several cellular processes, including metabolism, proliferation, cell survival, growth, and angiogenesis, marking it a critical target for CNI1-4 in the reversal of MDR in breast cancer.

### 2.3. GO Functional Annotation Analysis and KEGG Enrichment Analysis

KEGG enrichment and GO function annotation analyses were performed to identify the biological processes involved in MDR reversal in breast cancer by CNI1-4. The analysis revealed 109 biological processes (BPs), 22 cellular components (CCs), 35 molecular functions (MFs), and 142 KEGG pathways. The top 20 functional categories were ranked based on the number of annotations. Significantly upregulated BPs included phosphorylation, apoptosis regulation, cell proliferation, RNA polymerase II-mediated transcription, and DNA templating (Figure 4A). The most upregulated CCs comprised cytoplasmic, nuclear, plasma membrane, and nucleoplasmic components (Figure 4B), while the predominant MFs included protein binding, ATP binding, and protein kinase activity (Figure 4C). KEGG pathway analysis showed enrichment in the PI3K/AKT, MAPK, RAS, EGFR, and FoxO signaling pathways, with the PI3K-AKT pathway exhibiting the highest number of enriched targets. These findings align with prior research [27,28,29], demonstrating that the PI3K-AKT pathway facilitates MDR development by modulating ABC transporter expression. Importantly, AKT1 serves as a pivotal target in this process, as its upregulation is strongly associated with MDR progression. Building on this evidence, we hypothesize that CNI1-4 exerts its anti-MDR effects through a dual mechanism: (1) specific binding to AKT1, thereby suppressing the PI3K-AKT pathway activity, and (2) a subsequent reduction in P-gp-mediated drug efflux. Furthermore, this proposed mechanism is reinforced by CNI1-4’s additional interactions with complementary pathways, including MAPK and RAS, providing a more comprehensive explanation for its multidrug resistance reversal capability.

### 2.4. Construction of Compound–Target–Pathway Interaction Networks

To further investigate the molecular mechanisms by which CNI1-4 reverses MDR in breast cancer, a compound–target–pathway network was constructed (Figure 5). This network comprises 105 nodes and 567 pathways. The compounds were ranked by degree value in descending order: CNI3 (67), CNI1 (66), CNI4 (64), and CNI2 (57). KEGG pathways were similarly ranked, with the top ten being as follows: cancer pathway (29), PI3K-Akt signaling pathway (20), glycosaminoglycans in cancer (19), prostate cancer (17), MAPK signaling pathway (17), chemical carcinogenesis–receptor activation (16), lipids and atherosclerosis (16), RAS signaling pathway (16), human cytomegalovirus infection (15), and endocrine drug resistance (14). The degree value of 20 for the PI3K-Akt signaling pathway indicates that this pathway is involved in a greater number of interactions within the CNI1-4 network, suggesting that CNI1-4 may modulate PI3K-Akt signaling through its interactions with multiple targets, thereby exerting a significant influence on the pathway.

### 2.5. Molecular Docking

Preliminary virtual screening revealed that CNI1-4 exhibited strong binding affinity with P-gp, with substrate competition at the P-gp binding site identified as a key mechanism for overcoming MDR in breast cancer. Molecular docking analysis was conducted to assess the binding modes and potential interactions of CNI1-4 with the positive control drug VRP at the P-gp binding site (6FN1). The results, including interactions with specific amino acid residues, are presented in Table 1. The binding affinities of CNI1-4 with P-gp ranged from −9.6 to −10.2 kcal/mol, which were lower than the binding energy of CNI1-4 with VRP, indicating that CNI1-4 possesses potent anti-breast cancer MDR activity, with a binding strength surpassing that of the positive control. Additionally, molecular docking of CNI1-4 was performed against targets in the PPI network with degree values exceeding 40, including AKT1 (6S9W), ALB (8RGL), HSP90AA1 (2BZ5), EGFR (3POZ), CASP1 (6CKZ), ESR1 (3CBP), CCND1 (6P8E), SRC (2H8H), and PPARG (5Z6S). The docking results, shown in Figure 6, revealed binding energies for AKT1-CNI1-4, EGFR-CNI1-4, ESR1-CNI1-4, and SRC-CNI1-4 below −10 kcal/mol, indicating strong interactions. Notably, CNI1-4 exhibited the strongest binding with AKT1, consistent with network pharmacology findings.

Hydrogen bonding and hydrophobic interactions are the primary forces driving ligand–substrate binding. To further investigate the molecular mechanism by which CNI1-4 reverses MDR in breast cancer, the docking results of CNI1-4, VRP, and P-gp were visualized, with 3D and 2D binding diagrams presented in Figure 7A–E. CNI1, CNI2, CNI4, and VRP form hydrogen bonds with Tyr952, while CNI2 and CNI3 also interact with Tyr309. Additionally, CNI3 established hydrogen bonds with Gln989 and Gln346. Hydrophobic interactions occurred between CNI1, CNI2, CNI3, CNI4, and VRP with Leu64, Ile339, Met68, and Ala986, respectively. The binding sites of CNI1-4 and VRP at P-gp partially overlap, indicating that the reversal of MDR by CNI1-4 may involve competition with DOX for binding at a key P-gp site. The interactions between CNI1-4 and AKT1 are shown in Figure 7F–I. CNI2 and CNI3 form hydrogen bonds with Glu17, while CNI3 also interacts with Asp292, and CNI1 bonds with Arg273. Furthermore, CNI1, CNI2, CNI3, and CNI4 established hydrogen bonds with Leu210, Val270, Tyr272, Leu264, Asp274, Asn54, and Ile84, forming hydrophobic interactions at these sites. The molecular docking results demonstrate that CNI1-4 binds stably to the P-gp and AKT1 binding sites, specifically targeting residues Tyr952, Leu64, Ile339, and Ala986 at the P-gp interface, and Leu210, Val270, Glu17, and Asp292 at the AKT1 interface, displaying a strong binding affinity.

### 2.6. Molecular Dynamics Simulations

To validate the binding affinity of CNI1-4 to AKT1 and P-gp, molecular dynamics simulations were conducted on receptor–ligand complexes, with VRP serving as a control.

#### 2.6.1. MD Simulation of AKT1-CNI1-4 Complexes

The RMSD (root mean square deviation) curve illustrates the fluctuation of protein conformation [30]. As depicted in Figure 8A, AKT1-CNI1-4 stabilized after 100 ns, while AKT1-CNI2 maintained greater stability throughout the 100 ns period. This suggests that CNI1-4 binds to AKT1 with minimal conformational changes and a relatively stable interaction. The absence of disruptions in the RMSD curve throughout the simulation indicated that the protein structure remained stable, without any significant alterations, and the system as a whole was stable.

Rg (root mean square of radius) serves as an indicator of protein or complex compactness and conformational changes [31]. Lower Rg values, reflecting a more compact structure, typically correlate with enhanced stability, signifying well-folded proteins with intact internal interactions. As shown in Figure 8B, the AKT1-CNI1-4 complexes demonstrated stability throughout the 0–100 ns period, with the AKT1-CNI2 complex exhibiting the lowest Rg value, suggesting optimal stability.

In addition, SASA (solvent-accessible surface area) analysis was conducted to evaluate the solvent molecular accessibility of the surface regions of the complexes [32]. During the simulations, a slight reduction in the solvent-accessible surface area of AKT1-CNI1-4 at 100 ns was observed compared to the initial state, particularly for AKT1-CNI2, AKT1-CNI3, and AKT1-CNI4. This reduction indicates an enhancement in the binding stability of these complexes (Figure 8C).

#### 2.6.2. MD Simulation of P-gp-CNI1-4 and P-gp-VRP Complexes

Figure 9A presented the RMSD of P-gp-CNI1-4 and P-gp-VRP complexes. The RMSD curves remained relatively stable over the 0–500 ns period, with the P-gp-CNI1-4 complexes exhibiting less fluctuation compared to the P-gp-VRP complex, suggesting a more stable binding of the former. The Rg values for both P-gp-CNI1-4 and P-gp-VRP complexes were stable throughout the simulation, with the P-gp-CNI4 complex consistently showing a lower Rg value than the P-gp-VRP complex (Figure 9B). The SASA of the P-gp-CNI1-4 complexes gradually decreased, indicating an increase in binding strength, whereas the SASA of the P-gp-VRP complex remained largely unchanged (Figure 9C). During the 0–500 ns period, the number of hydrogen bonds in the P-gp-VRP complexes ranged from 0 to 26, consistently higher than those in the P-gp-CNI1-4 complexes (Figure 9D).

MD simulations revealed that both the P-gp-CNI1-4 and P-gp-VRP complex structures underwent changes before stabilizing.

### 2.7. Binding Free Energy MM-PBSA Calculations

The MM-PBSA method effectively evaluates the binding free energy of ligand–receptor complexes, offering detailed insights into binding affinities and molecular interactions. Binding free energy calculations were performed using the final stable 50 ns RMSD trajectory, with contributions analyzed across several factors: molecular mechanical energy (including van der Waals and electrostatic interactions), polarized solvation energy, and unpolarized solvation energy (solvent-accessible surface area, SASA).

#### 2.7.1. Calculation of the Binding Free Energy of AKT1-CNI1-4

The binding affinities of the ligands were calculated using the g-mmpbsa method. As presented in Table 2, the total binding free energies for AKT1-CNI1, AKT1-CNI2, AKT1-CNI3, and AKT1-CNI4 were −26.6162, −42.1028, −32.4338, and −34.7798 kJ/mol, respectively. The binding free energies of AKT1-CNI2, AKT1-CNI3, and AKT1-CNI4, all below −30 kJ/mol, indicate strong interactions upon binding.

#### 2.7.2. P-gp-CNI1-4 and P-gp-VRP Binding Free Energy Calculations

In MM-PBSA calculations, the total binding free energies of P-gp-CNI1, P-gp-CNI2, P-gp-CNI3, P-gp-CNI4, and P-gp-VRP were −38.5902, −39.0606, −35.9709, −45.4900, and −41.7985 kcal/mol, respectively (Table 3). All complexes exhibited strong interactions, with P-gp-CNI4 displaying the lowest binding free energy, stronger than that of P-gp-VRP. However, cellular experiments revealed that CNI3 had the most effective reversal of MDR in MCF-7/DOX-resistant cells when combined with DOX, while CNI4 alone demonstrated the greatest cytotoxicity against these cells. The discrepancy between the computational and experimental results can be attributed to the fact that the cellular experiments involved the co-administration of CNI1-4 with DOX, whereas molecular dynamics simulations only examined the interaction of CNI1-4 with P-gp. This difference arises from the divergence between the simulated and experimental environments. Future simulations involving both CNI1-4 and DOX in conjunction with P-gp will be conducted to further validate these findings.

### 2.8. Discussion

This study identified CNI1-4 as potential candidate compounds for reversing breast cancer multidrug resistance (MDR) using computational biology approaches. Our results suggest that CNI1-4 may exert its effects by dual mechanisms: (1) targeting AKT1 in the PI3K-AKT pathway and (2) inhibiting P-gp-mediated drug efflux. These findings align with previous studies, which demonstrated that PI3K-AKT pathway activation promotes MDR via ABC transporter upregulation. Notably, our work complements the research by Gu and Huang et al. [27], who showed that AKT1 inhibition improves the chemotherapy response in drug-resistant cells. However, while their study focused on synthetic AKT inhibitors, our natural compound CNI1-4 exhibits comparable mechanisms with potentially improved safety profiles. Compared to verapamil (a standard P-gp inhibitor), CNI1-4 demonstrated superior binding energy and stability in docking and molecular dynamics (MD) simulations, suggesting structural advantages for overcoming specificity limitations. Furthermore, network pharmacology analysis revealed that CNI1-4 modulates multiple targets (e.g., the concurrent regulation of the MAPK/RAS pathway), which may explain its enhanced efficacy over single-target inhibitors.

However, a key limitation of this study is the lack of experimental validation. Future research should focus on the following: in vitro validation using Western blot and rhodamine-123 efflux assays to assess P-gp expression and activity in MCF-7/DOX cells; in vivo models, including pharmacokinetic studies and MDR reversal evaluation in xenograft mice; the structural optimization of CNI1-4 using MD simulation data to enhance binding affinity.

Despite these limitations, our integrated computational strategy provides a robust framework for elucidating CNI1-4’s mechanisms. The predicted AKT1-P-gp regulatory axis offers testable hypotheses for future research, accelerating the development of next-generation MDR inhibitors.

## 3. Materials and Methods

### 3.1. Analysis of CNI1-4 Drug Targets Using Network Pharmacology

Pharmmapper (https://www.lilab-ecust.cn/pharmmapper/ (accessed on 5 May 2024)) and SwissTargetPrediction (https://swisstargetprediction.ch/ (accessed on 5 May 2024)) are online platforms designed for predicting the potential targets of compounds based on their molecular structures. In this study, the compounds CNI1-4 were analyzed to identify their potential targets using these two databases. Parameters were set as Norm fit > 0.7 and Probability > 0 to refine the selection. After retrieving the target data for CNI1-4, gene normalization was performed using the UniProt database (https://www.uniprot.org/ (accessed on 6 May 2024)) to verify the accuracy and consistency of the target information. Ultimately, four potential targets for CNI1-4 were identified.

### 3.2. Network Pharmacology Screening for MDR Targets in Breast Cancer

The Genecards database, which integrates information on approximately 150 genes [33], and the OMIM database, a comprehensive resource for human genes and genetic diseases [34], were searched to identify targets associated with MDR in breast cancer. Using the keywords “breast cancer” and “multidrug resistance,” relevant data were retrieved from both Genecards (https://www.genecards.org/ (accessed on 11 May 2024)) and OMIM (https://www.omim.org/ (accessed on 11 May 2024)).

### 3.3. Potential Targets for the Intersection of Active Compounds CNI1-4 with MDR in Breast Cancer

The intersection analysis of CNI1-4 targets and breast cancer MDR-related targets was conducted using the Venny 2.1 online tool (https://bioinfogp.cnb.csic.es/tools/venny/ (accessed on 14 May 2024)) to identify common target genes, which were selected as potential candidates for CNI1-4 in reversing breast cancer MDR.

### 3.4. Construction of Protein–Protein Interaction (PPI) Networks

To further explore the molecular mechanisms underlying CNI1-4’s role in reversing MDR in breast cancer, the potential targets of CNI1-4 were submitted to the STRING database (https://cn.string-db.org/ (accessed on 16 May 2024)) for analysis. Data were imported by selecting “Homo sapiens” as the species and applying a combined score threshold of >0.7. Cytoscape 3.9.0 was then employed to construct the protein–protein interaction (PPI) network. The final PPI network was generated through CentiScape analysis, filtering core targets based on topological parameters, including degree, betweenness, and closeness, and selecting those exceeding the median value in all three measures.

### 3.5. GO and KEGG Enrichment Analyses

GO enrichment analysis categorizes genes by molecular function (MF), biological process (BP), and cellular component (CC) [35], while KEGG enrichment focuses on identifying gene associations with cellular signaling pathways. The GO and KEGG enrichment of potential target genes involved in reversing MDR in breast cancer by CNI1-4 was analyzed using the DAVID database (https://david.ncifcrf.gov/ (accessed on 18 May 2024)). To pinpoint key pathways, data were filtered with thresholds of FDR < 0.05 and *p* < 0.05 and subsequently visualized and analyzed using the Microbiotics Online Bioinformatics Analysis Platform (https://bioinformatics.com.cn/ (accessed on 19 May 2024)).

### 3.6. Molecular Docking

This study aimed to predict potential interactions between CNI1-4 and target proteins identified through network pharmacology, as well as interactions between CNI1-4, VRP, and P-gp. The molecular structure of CNI1-4 was constructed in ChemDraw 20.0 and optimized to the lowest energy conformation. The three-dimensional structure of VRP was retrieved from the PubChem database (https://pubchem.ncbi.nlm.nih.gov/ (accessed on 16 June 2024)). The 3D structures of the selected targets were obtained from the PDB database (https://www.rcsb.org/ (accessed on 16 June 2024)) and saved in PDB format. Using PyMOL 2.5.5 software, small-molecule ligands and water molecules were removed from the protein structures, and the data were imported into AutodockTools 1.5.7 for hydrogenation, with the addition of Kollman charges, and conversion to pdbqt format. Molecular docking was conducted with Autodock Vina 1.1.2, incorporating pocket information (refer to Appendix A). Heat maps were generated based on docking results, and complexes exhibiting optimal binding affinity were visualized and analyzed with PyMOL 2.5.5 and Discovery Studio 2019 software.

### 3.7. Molecular Dynamics (MD) Simulations

Molecular dynamics simulations were conducted using a GPU-accelerated version of AMBER22 (pmemd.cuda) on the compounds with the highest binding energies from molecular docking results, complexed with receptor proteins [36], to assess their binding modes and energies. The simulations employed the ff19SB protein force field [37] and the TIP3P water model [38], with the restrained electrostatic potential (RESP) charges of small molecules derived from a previous study [39]. The AMBER GAFF2 force field [40] was applied for other parameters. The initial structures from molecular docking were placed in a rectangular water box, followed by energy minimization using the conjugate gradient algorithm. Subsequently, the systems were gradually heated over 50 ps from 0 to 300 K in an NVT ensemble, then subjected to 500 ns and 100 ns molecular dynamics simulations in an isothermal isobaric system synthesis (NPT) ensemble. Temperature control was maintained at 300 K using a Langevin thermostat, and pressure was regulated at 1.0 bar via a Monte Carlo constant pressure algorithm. A 2 fs integration step was used, with trajectory snapshots saved every 100 ps for further analysis.

### 3.8. Combined Free Energy Calculations

Free energy calculations provide valuable insight into the strength of intermolecular interactions within receptor–ligand complexes. The MM-PBSA (molecular mechanics Poisson–Boltzmann surface area) method integrates molecular mechanics (MM) with electrolyte solution (PBSA) modeling to efficiently assess the free energy variations in molecules in solution [41]. The MMPBSA.py tool in AMBER22 was employed to compute the binding affinity of receptor–ligand complexes, shedding light on the stability and interaction mechanisms between CNI1-4 compounds and key targets.

## 4. Conclusions

This study systematically investigates the key targets and potential molecular mechanisms underlying the reversal of MDR in breast cancer by CNI1-4, employing network pharmacology, molecular docking, and molecular dynamics simulations. The results indicate that CNI1-4 significantly enhances the sensitivity of breast cancer cells to chemotherapeutic agents by targeting P-gp and modulating the PI3K/Akt signaling pathway. Further molecular docking and dynamics simulations elucidate the binding modes and stability of CNI1-4 with P-gp and AKT1, providing a theoretical framework for its application as an MDR reversal agent in breast cancer.

This study demonstrates the complex molecular mechanism by which CNI1-4 reverses MDR in breast cancer through multi-dimensional network pharmacology and computational simulation analyses, offering a novel theoretical foundation for future drug development and clinical applications.

## Figures and Tables

**Figure 1 ijms-26-03813-f001:**
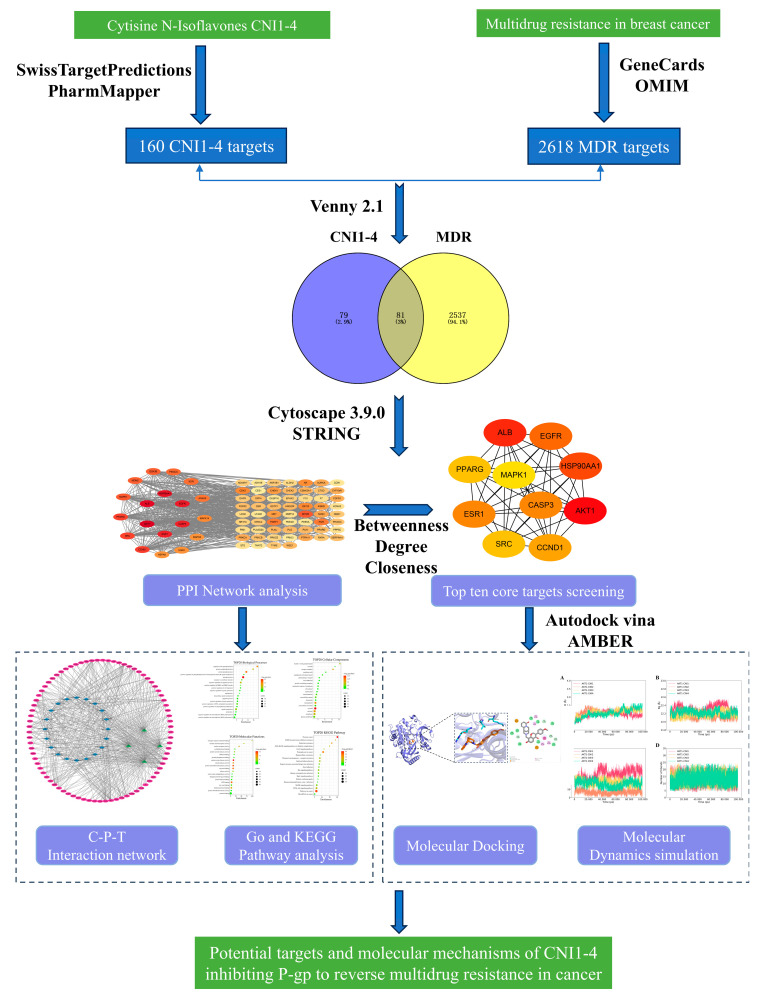
Flowchart of the CNI1-4 inhibition of P-gp to Reverse Multidrug Resistance In Breast Cancer Study.

**Figure 2 ijms-26-03813-f002:**
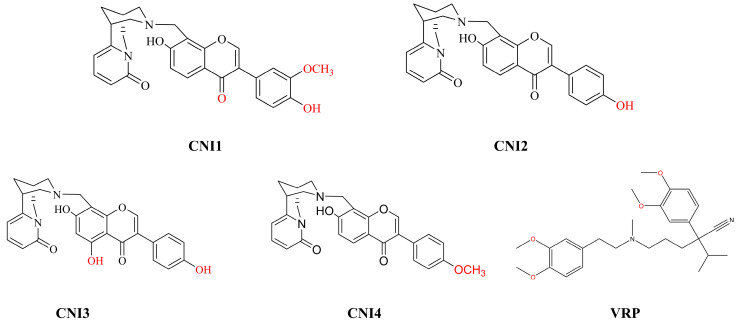
Structural formulae of CNI1-4 compounds and positive control drugs.Red font indicates the modified positions on the benzene rings.

**Figure 3 ijms-26-03813-f003:**
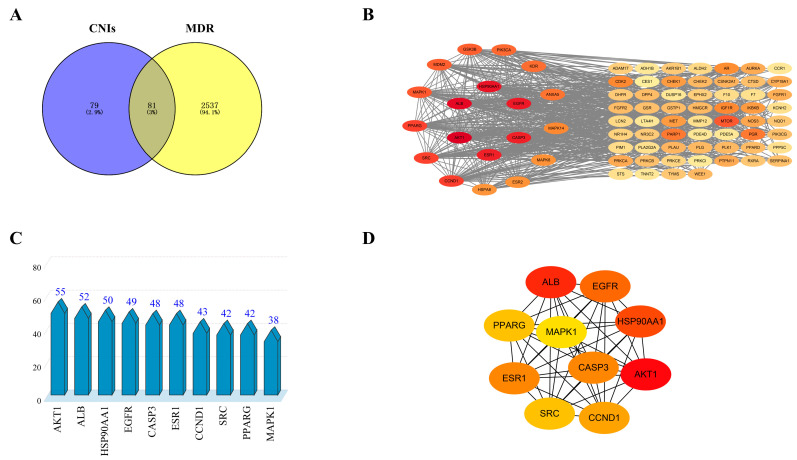
CNI1-4 reversal of MDR potential targets and protein–protein interaction (PPI) networks in breast cancer. (**A**) Venn diagram showing potential targets. (**B**) PPI network of 81 potential targets. (**C**) Top 10 potential targets sorted by degree value. (**D**) PPI network of core targets in the top 10 of the ranking.

**Figure 4 ijms-26-03813-f004:**
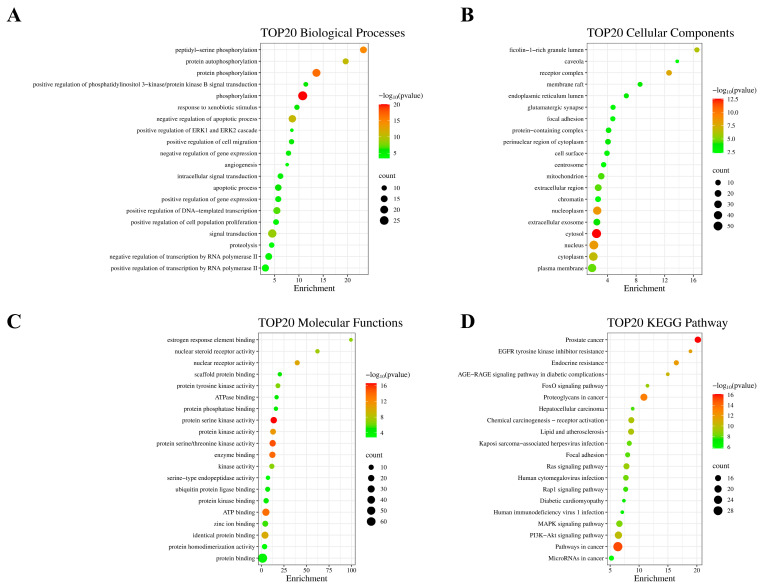
Functional annotation and KEGG pathway enrichment analysis of 81 potential breast cancer MDR targets. (**A**) Bubble plot showing the top 20 BPs. (**B**) Bubble plot showing the top 20 CCs. (**C**) Bubble plot showing the top 20 MFs. (**D**) Bubble plot showing the top 20 KEGG pathways.

**Figure 5 ijms-26-03813-f005:**
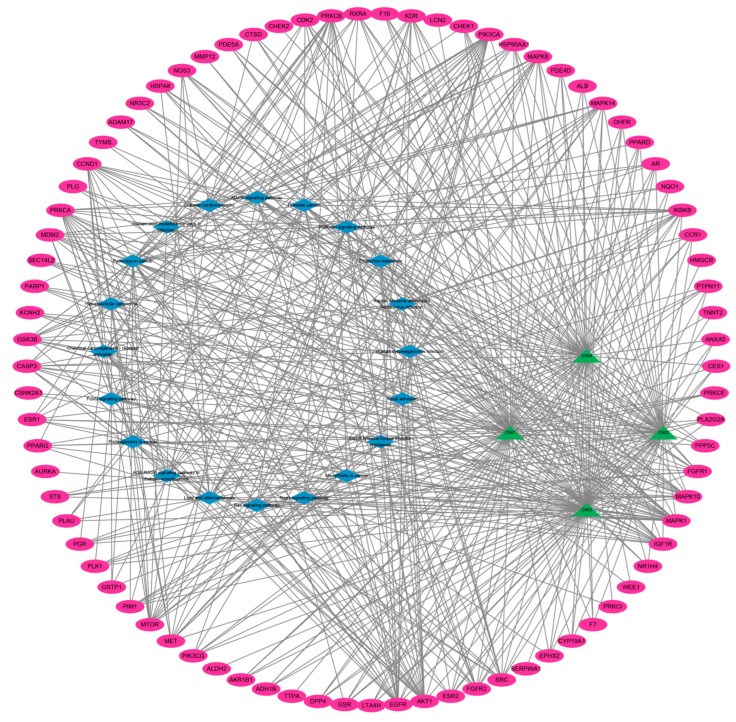
Compound–target–pathway interaction network. Pink represents the target, green represents CNI1-4, and blue represents the signaling pathway.

**Figure 6 ijms-26-03813-f006:**
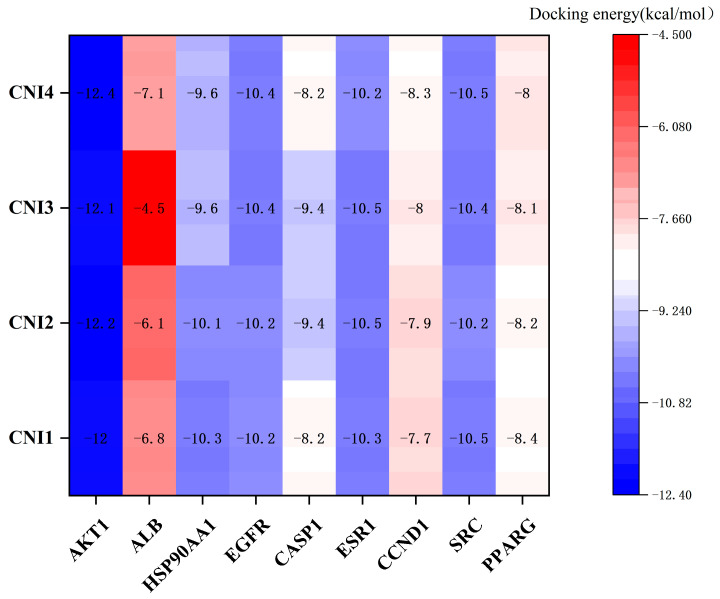
Docking scores of CNI1-4 against targets with degree values > 40. A lower score indicates a stronger binding ability.

**Figure 7 ijms-26-03813-f007:**
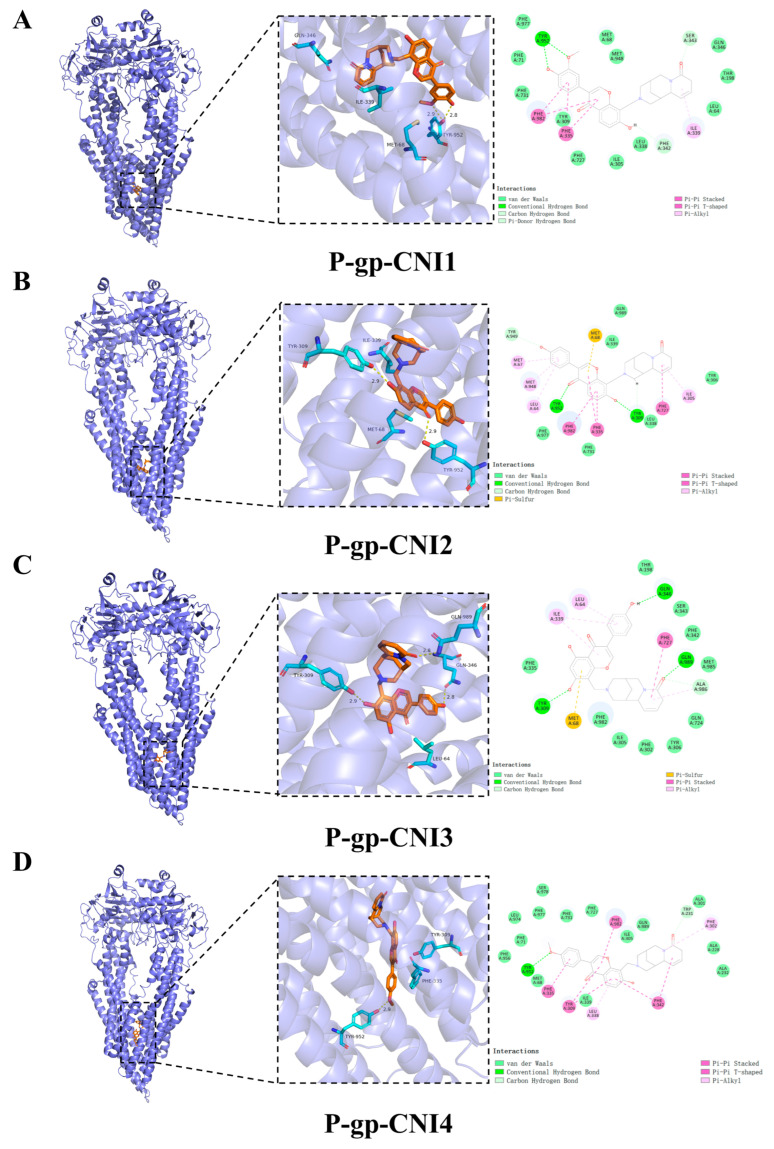
Visualization and analysis of CNI1-4, VRP, P-gp, and AKT1 molecular docking 2D and 3D images. Small-molecule compounds are represented as brown sticks in the 3D structure. The key residues are represented as cyan sticks, respectively. Hydrogen bonds are indicated as yellow dashed lines. (**A**) P-gp-CNI1 complex. (**B**) P-gp-CNI2 complex. (**C**) P-gp-CNI3 complex. (**D**) P-gp-CNI4 complex. (**E**) P-gp-VRP complex. (**F**) AKT1-CNI1 complex. (**G**) AKT1-CNI2 complex. (**H**) AKT1-CNI3 complex. (**I**) AKT1-CNI4 complex.

**Figure 8 ijms-26-03813-f008:**
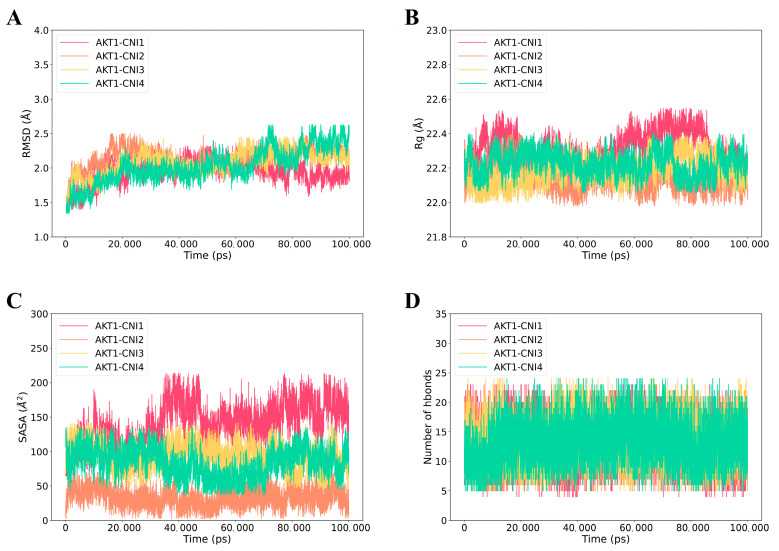
Molecular dynamics (MD) simulations of the AKT1-CNI1-4 complex. (**A**) Root mean square fluctuation (RMSD) values extracted from the protein fit ligand of the protein–ligand docked complexes. (**B**) The compactness of the protein structure in terms of the radius of gyration (Rg). (**C**) Solvent-accessible surface area (SASA) analysis. (**D**) H bond formation between AKT1 and CNI1-4.

**Figure 9 ijms-26-03813-f009:**
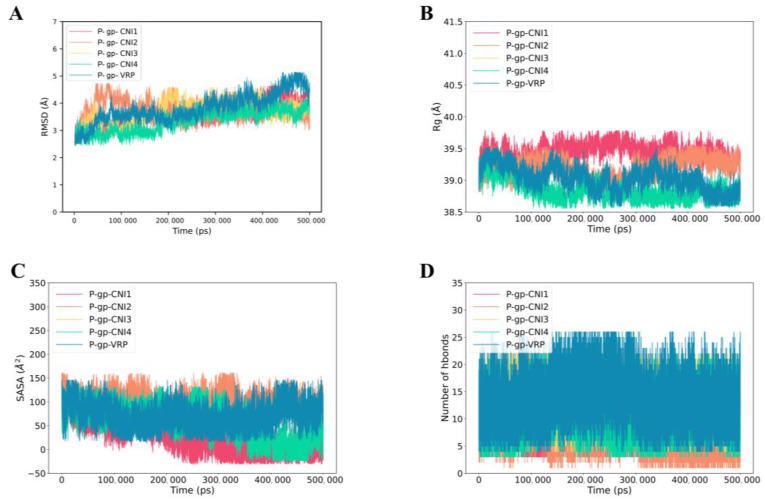
Molecular dynamics (MD) simulations of P-gp-CNI1-4 and P-gp-VRP complexes. (**A**) Root mean square fluctuation (RMSD) values extracted from the protein fit ligand of the protein–ligand docked complexes. (**B**) The compactness of the protein structure in terms of the radius of gyration (Rg). (**C**) Solvent-accessible surface area (SASA) analysis. (**D**) H bond formation between P-gp and CNI1-4, VRP.

**Table 1 ijms-26-03813-t001:** Docking results of CNI1-4 and VRP with P-gp.

Compound	Docking Score (Kcal/mol)	Hydrogen Bonding	Hydrophobic Interactions
CNI1	−10.4	Tyr952	MET68,Phe71,Ile305, Tyr309,Phe335,Leu338, Ile339,Phe342,Ser343, Gln346,Phe731,Phe982
CNI2	−10.1	Tyr309, Tyr952	Leu64,Met67,Met68, Ile305,Phe335,Phe727, Met948,Tyr949,Phe982
CNI3	−10.0	Gln346, Tyr309, Gln989	Leu64,Met68,Ile339, Ala986
CNI4	−10.2	Tyr952	Phe71,Trp231,Ala301, Phe302,Ile305,Tyr309, Phe335,Leu338,Ile339, Phe342,Phe731,Phe982, Gln989
VRP	−7.9	Tyr952, Gln989	Met68,Ile305,Tye309, Phe335,Leu338,Phe342, Gln724,Phe727,Met948, Phe982,Ala986

**Table 2 ijms-26-03813-t002:** Analysis of protein ligand AKT1-CNI1-4 molecular mechanics/generalized born surface area (MMPBSA) (KJ/mol).

	CNI1	CNI2	CNI3	CNI4
VDWAALS	−51.0246	−70.7361	−59.6757	−61.6645
EEL	−4.4372	−12.4815	−0.6747	−4.5164
EPB	34.2307	46.2531	33.3708	36.9572
DELTA G gas	−55.4618	−83.2176	−60.3504	−66.1808
DELTA G solv	28.8456	41.1148	27.9166	31.4011
DELTA TOTAL	−26.6162	−42.1028	−32.4338	−34.7798

**Table 3 ijms-26-03813-t003:** Analysis of protein ligand P-gp-CNI1-4, P-gp-VRP molecular mechanics/generalized born surface area (MMPBSA) (KJ/mol).

	CNI1	CNI2	CNI3	CNI4	VRP
VDWAALS	−63.2064	−56.8462	−58.2130	−66.5229	−58.8949
EEL	−7.5823	−1.8142	−2.4056	−3.0904	−8.3045
EPB	37.7542	24.8517	29.9995	29.6317	31.6844
DELTA G gas	−70.7888	−58.6568	−60.6186	−69.6133	−67.1994
DELTA G solv	32.1985	19.5962	24.6478	24.1233	25.4008
DELTA TOTAL	−38.5902	−39.0606	−35.9709	−45.4900	−41.7985

## Data Availability

The original contributions presented in the study are included in the article/Appendix A. Further inquiries can be directed to the corresponding authors.

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
