# Peer review of "Molecular Mechanisms of Reversal of Multidrug Resistance in Breast Cancer by Inhibition of P-gp by Cytisine N-Isoflavones Derivatives Explored Through Network Pharmacology, Molecular Docking, and Molecular Dynamics"

_ijms, 2025, doi:10.3390/ijms26083813_

Round 1
Reviewer 1 Report
Comments and Suggestions for Authors
The study investigates the molecular mechanisms through which N-isoflavonoid derivatives of chrysophyllin (CNI1-4) reverse multidrug resistance (MDR) in breast cancer by P-glycoprotein (P-gp) inhibition. Using a comprehensive strategy integrating network pharmacology, molecular docking, and molecular dynamics simulations, the authors screened 81 potential targets associated with MDR and assessed their suitability. They determined that CNI1-4 compounds have high binding capacity to P-gp, AKT1, and SRC, which are targeted in PI3K-Akt and MAPK signaling pathways. The simulations confirmed that CNI1-4 compounds stabilize their binding to P-gp, making them more valuable as P-gp inhibitors. The study provides beneficial data for the development of novel P-gp inhibitors that can overcome MDR in breast cancer.
This research is, in my view, substantially significant because it identifies a novel strategy in overcoming multidrug resistance (MDR) in breast cancer, which is a major obstacle in chemotherapy. The discovery of CNI1-4 as promising P-glycoprotein inhibitors provides new information for the enhancement of treatment effectiveness. Besides, the application of integrated methods, i.e., network pharmacology and molecular dynamics simulations, enhances the completeness of the findings and qualifies the study as a solid basis for future research on drug development against MDR.
My opinion is that the article can be published if some very important issues are addressed.
- Introduction discusses the issue of MDR in breast cancer and the need for effective P-gp inhibitors very thoroughly. Redundancy exists, however, such as drug ineffectiveness and the role of P-gp in drug resistance. It would be more cohesive to introduce more briefly the problem statement and highlight more the contribution of the present study.
-The findings are described comprehensively, though the description of the molecular pathways involved (PI3K-Akt, MAPK) may be more consistent. The data presentation from GO and KEGG enrichment analyses is lengthy, but their alignment with the role of CNI1-4 in cancer cell lines may be described more clearly.
- The findings are described as the focal discussion point, yet the interpretation by comparison with previous studies is not put forward forcefully. It would be beneficial to draw more express comparison with previous studies and to discuss more comprehensively how the new data adds to the understanding of the reversal of MDR in breast cancer.
-Even though the research makes use of sound computational techniques, not having experimental data derived from in vitro and in vivo studies lowers the direct relevance of the outcomes. Getting over such shortcomings and specifying future opportunities for confirmation of the findings using laboratory experiments would reinforce the study.
-The tables and charts contain a wealth of information, though some of it is given in a condensed manner, which may be challenging to readers who lack knowledge of the subject. A short commentary next to each chart and table aimed at pointing out the important information would strengthen the presentation of data.
Reviewer 2 Report
Comments and Suggestions for Authors
Compounds CNI1-CNI4 (previously reported by team of Zhicheng Zuo, ref. 19) were classified as cytisine N-isoflavones. The following names are used for these compounds:
- chrysophylline derivatives (what does chrysophylline mean?),
- N-isoflavone derivatives of chrysoerioline (what does chrysoerioline mean?),
- chrysoerioline-isoflavone derivatives (see above),
- chrysoeriol N-isoflavonoid derivatives (chrysoeriol is a flavone),
- chrysin-isoflavone-based compounds (chrysin is a flavone),
- chrysin N -isoflavonoid derivatives, or
- chrysoidine-isoflavone derivatives (chrysoidine is 3-amino-4-phenylazo-phenyl)amine).
Unfortunately, all the abovementioned names are wrong.
The following names mentioned in the manuscript are also wrong: chrysoerioline N-methylene-(4',7-dihydroxy-3'-methoxy)-isoflavone (what does chrysoerioline mean?) and chrysin N-methylene-(4',7-dihydroxy-3'-methoxy)-isoflavone (chrysin is a flavone).
The use of such great number of manes in the text (apart from the fact that all of them are wrong) is confusing.
The presented study is strictly theoretical. Interactions of the discussed compounds with the identified target proteins need to be verified. For this reason, the presented study can be considered as a preliminary stage of detail in vitro assays.
Comments on the Quality of English LanguageSome phrases should be modified because they sound illogically. For instance p. 3, line 98, “This study investigates the molecular mechanism by which CNI1-4 reverses MDR in breast cancer using network pharmacology, .......”
Round 2
Reviewer 1 Report
Comments and Suggestions for Authors
The authors have revised the manuscript in accordance with the suggestions provided. I recommend the publication of this article.
Reviewer 2 Report
Comments and Suggestions for Authors
The Authors reasonably explained the reviewer's doubts.
Please note that the "N" descriptor in "cytisine N-isoflavones" must be written in italics.